# LncRNAs as Regulators of Atherosclerotic Plaque Stability

**DOI:** 10.3390/cells12141832

**Published:** 2023-07-12

**Authors:** Aleksa Petkovic, Sanja Erceg, Jelena Munjas, Ana Ninic, Sandra Vladimirov, Aleksandar Davidovic, Luka Vukmirovic, Marko Milanov, Dane Cvijanovic, Tijana Mitic, Miron Sopic

**Affiliations:** 1Clinical-Hospital Centre “Dr Dragiša Mišović-Dedinje”, 11000 Belgrade, Serbia; aleksa93@gmail.com; 2Department of Medical Biochemistry, Faculty of Pharmacy, University of Belgrade, 11000 Belgrade, Serbia; sanja.erceg@pharmacy.bg.ac.rs (S.E.); miron.sopic@pharmacy.bg.ac.rs (M.S.); 3Intern Clinic, Clinical Ward for Cardiovascular Diseases, Clinical-Hospital Centre Zvezdara, 11000 Belgrade, Serbia; 4Department for Internal Medicine, Faculty of Dentistry, University of Belgrade, 11000 Belgrade, Serbia; 5Centre for Cardiovascular Science, The Queen’s Medical Research Institute, Edinburgh EH16 4TJ, UK

**Keywords:** long non-coding RNAs, atherosclerosis, plaque stability

## Abstract

Current clinical data show that, despite constant efforts to develop novel therapies and clinical approaches, atherosclerotic cardiovascular diseases (ASCVD) are still one of the leading causes of death worldwide. Advanced and unstable atherosclerotic plaques most often trigger acute coronary events that can lead to fatal outcomes. However, despite the fact that different plaque phenotypes may require different treatments, current approaches to prognosis, diagnosis, and classification of acute coronary syndrome do not consider the diversity of plaque phenotypes. Long non-coding RNAs (lncRNAs) represent an important class of molecules that are implicated in epigenetic control of numerous cellular processes. Here we review the latest knowledge about lncRNAs’ influence on plaque development and stability through regulation of immune response, lipid metabolism, extracellular matrix remodelling, endothelial cell function, and vascular smooth muscle function, with special emphasis on pro-atherogenic and anti-atherogenic lncRNA functions. In addition, we present current challenges in the research of lncRNAs’ role in atherosclerosis and translation of the findings from animal models to humans. Finally, we present the directions for future lncRNA-oriented research, which may ultimately result in patient-oriented therapeutic strategies for ASCVD.

## 1. Introduction

Cardiovascular diseases (CVDs) remain the leading cause of morbidity and mortality across the globe [1]. It is currently estimated that 19.1 million people die from CVDs each year, with an increase of 18.71% of deaths caused by CVDs between 2010 to 2020 [1], which accounts for 32% of deaths worldwide [2]. Ischaemic heart disease (IHD), caused by atherosclerotic plaque occlusion of the major coronary blood vessels, underlie all CVDs. Acute coronary syndromes (ACS) and sudden death account for most IHD-related deaths (1.8 million deaths worldwide) [3]. The ACS most often results from a rupture or erosion of the culprit atherosclerotic plaque within the vessel wall, causing coronary thrombosis and blockage of blood flow to the heart [4]. Less frequently, ACS is a consequence of epicardial or microvascular spasm [5]. Which mechanism will be responsible for the development of ACS depends on the type of plaque, its stability and its interaction with the microvascular environment of the patient [4,5,6].

Although it is known that atherogenesis starts with the activation of endothelial cells of the blood vessel wall [7] induced by stimuli such as angiotensin II, inflammation, or oxidative stress [8], it is still unknown what is the first event in the cascade leading to downstream events. Nevertheless, the entry of low-density lipoproteins (LDL) and triglyceride-rich lipoproteins in the blood vessel wall occurs readily, together with the entrance of several classes of leukocytes into the sub-endothelial intima layer [8]. Both chemical and physical modifications of the lipoproteins take place [9], inducing proinflammatory changes in the surrounding cells [10]. Macrophages, the most numerous cells that populate plaques, ref. [11], take up modified lipoproteins, leading to the formation of foam cells and enlargement of the tunica intima [12]. Vascular smooth muscle cells (VSMCs) of the tunica media migrate into the endothelial intima [13], proliferate, release chemoattractants, and form a substantial fraction of lipid-laden foam cells [14]. They represent the main cellular fraction and are responsible for the synthesis of the strength-giving extracellular matrix components of the fibrous cap [15]. Furthermore, in the growing atherosclerotic plaque the number of dead and apoptotic cells is increasing. Such a microenvironment within the plaque leads to impairment of efferocytosis [16]; namely, damaged cells are not being recognised and cleared, leading to the accumulation of foam cells, necrosis, and further propagation of inflammation. Traditionally, two main classes of high-risk plaques are distinguished: rupture-prone and erosion-prone plaques. Rupture-prone plaques are thin-cap fibroatheromas (TCFA) with a large lipid-rich necrotic core, infiltrated with macrophages and inflammatory cells, characterized by neovascularization, microcalcification, and intraplaque haemorrhage, and covered by a thin fibrous cap [6]. Erosion-prone plaques are more heterogeneous and lack a distinct central lipid rich core; they contain abundant proteoglycans and glycosaminoglycans as well as SMCs [6,17]. The mechanism of superficial erosion does not involve inflammation mediated by macrophages, but mainly by neutrophils [18]. Due to relatively good control measures such as smoking bans and the prolonged use of lipid lowering and antihypertensive drugs, most frequent rupture-prone plaques are shown to be declining, whereas erosion-prone plaques currently appear to be on the rise [19,20]. However, different plaque phenotypes could require different types of treatment, hence future approaches for the prognosis, diagnosis, and classification of ACS need to embrace and reflect the diverse complex mechanisms that lie behind acute ischemia [21]. Therefore, novel tools are needed to distinguish between different plaque phenotypes and identify patients at high risk for acute events are needed [21].

It has been documented that non-coding RNAs (ncRNAs) drive crucial processes in the atherosclerotic plaque development, and regulate the expression of various proteins involved in plaque progression and its vulnerability [22,23]. Reports from the ENCODE (Encyclopaedia of DNA Elements) project reveals that up to 80% of the human genome has the capacity to transcribe into non-coding RNAs (ncRNAs) [24]. Among them, long ncRNAs (lncRNAs) are the most prevalent and functionally diverse class [25]. LncRNAs are transcribed into transcripts that can be classified as: long intergenic non-coding (linc)-RNAs, long non-coding antisense transcripts, long intronic non-coding RNAs, and non-overlapping antisense lncRNAs [26]. They affect every level of gene regulation including epigenetic, transcriptional, post-transcriptional, translation regulation, and post-translational modification. Interestingly lncRNAs by definition do not code proteins and typically lack functional initiation and termination codons, but some lncRNAs have been shown to translate into micropeptides [27,28].

To date, lncRNAs have been shown to influence plaque development by regulating immune responses, lipid metabolism, extracellular matrix remodelling, endothelial cell function, and vascular smooth muscle function, sometimes affecting multiple molecular signalling pathways simultaneously [22,23,29]. Moreover, extracellular lncRNAs are relatively resistant to degradation and can be detected in various biological fluids, making them attractive biomarker candidates. In this review, we present a summary of current knowledge on the links between lncRNAs and plaque development and stability, recent advances, challenges, and potential future directions in the field.

## 2. Long Non-Coding Ribonucleic Acids (lncRNAs) Implicated in Plaque Instability

Atherosclerotic plaque represents a dynamic environment where the interplay between various cell types, including endothelial cells, immune cells, and VSMCs, governs the plaque phenotype and its vulnerability. In the subsequent sections, we will discuss a selection of previously studied long non-coding ribonucleic acids (lncRNAs) that have been implicated in regulating key functions of these cells (Figure 1), exerting both pro- and anti-atherogenic effects that contribute, directly or indirectly, to plaque stability. It is worth mentioning that some of the lncRNAs presented in this review, such as *NEAT1* and *MALAT1*, are constitutively expressed in different tissues, indicating their potential involvement in broader biological processes. However, despite their widespread expression, our focus within this review will primarily be on investigating the functions and regulatory roles of these lncRNAs specifically within the cells that drive progression of atherosclerosis. We will discuss their specific roles and implications within the context of atherosclerotic plaque development, highlighting their relevance to the pathogenesis and progression of ASCVD (Table 1).

### 2.1. Macrophages

As mentioned earlier, the key events in the development of atherosclerosis revolve around macrophages, their transition into subendothelial space and transformation into foam cells [30]. The role of macrophages in the subendothelium is quite complex and as such is interconnected with the processes of deregulated cholesterol transport, the reduced ability of macrophages to perform apoptosis, phagocytosis, efferocytosis, and the presence of inflammation and oxidative stress [31].

### 2.2. Cholesterol Transport

Cholesterol build-up in the macrophages can be aborted by the efficient reverse cholesterol transport mediated by the HDL. However, this process is often dysregulated during the development of atherosclerosis [30]. Currently, a number of lncRNAs are found to be implicated in this process. A recent study by Yu et al. [32] showed that overexpression of the lncRNA Kcnq1 overlapping transcript 1 (*KCNQ1OT1*) in apoE influences the efficiency of RCT. This occurs through the competitive binding of *KCNQ1OT1* to miR-452-3p, resulting in increased *HDAC3* expression and decreased *ABCA1* expression in macrophages. In contrast, the knockdown of *KCNQ1OT1* prevented atherosclerosis in −/− mice thanks to reduced lipid accumulation in THP-1 cells [26]. On the other hand, the lncRNA *MeXis* shows a different effect on RTC in macrophages [33]. Namely, *MeXis* potentiates LXR-dependent transcription of *ABCA1* through loosened actions of the transcriptional coactivator DDX17, which has been shown to be enriched at LXR-binding sites in *ABCA1* enhancer regions. Reduced expression of *ABCA1* was found in the bone marrow of Ldlr−/− mice after transplantation of *MeXis−/−* hematopoietic cells, compared to expression in the bone marrow of mice transplanted with wild-type (WT) hematopoietic cells. In addition, mice with transplanted *MeXis−/−* blood cells have a significantly increased plaque burden after 17 weeks of the Western diet. Another lncRNA with LXR-regulating function is *CHROME* [34]. *CHROME* expression is tightly controlled by dietary and cellular cholesterol levels. Namely, an increase in cholesterol levels can activate the sterol-activated transcription factor LXR, leading to the upregulation of *CHROME*. Additionally, in THP-1 macrophage cells, this lncRNA expresses its cytosolic functions by sequestering miR-27b, miR-33a, miR-33b, and miR-128 from their target mRNAs. Loss of function of these miRNAs could significantly increase the mRNA levels of *ABCA1*, *OSBPL6* and *NPC1*, which is an important step in maintaining the efflux of excess cellular cholesterol, especially in macrophages in arterial walls. *CHROME* expression levels were found to be higher in cholesterol-laden macrophages in vitro and in human atherosclerotic plaques, while CHROME-7WT-expressing THP-1 macrophages showed lower cytoplasmic accumulation of lipid droplets [34]. Another lncRNA associated with the regulation of *ABCA1* expression is Prostate Cancer Antigen 3 (*PCA3*). Its downregulation has been associated with foam cell formation and the development of atherosclerosis [35]. More specifically, the atheroprotective effect of this lncRNA is achieved through the PCA3/miR-140-5p/RFX7/ABCA1 axis. *PCA3* sponges miR-140-5p, which upregulates *RFX7*, one of the DNA binding proteins. *RFX7*, in turn, upregulates the expression of *ABCA1* in foam cells of *ApoE*−/− mice. This leads to increased cholesterol efflux and decreased lipid accumulation in macrophages [35]. On the other hand, the lncRNA *GAS5* showed significant upregulation in the AS mouse model and in ox-LDL-treated macrophages [36]. This lncRNA acts as a suppressor of miR-135a, which has been shown to have potent anti-atherosclerotic effects. In macrophages treated with ox-LDL, restoration of miR-135a expression reversed the effects of GAS5—particularly those related to dysregulation of lipid metabolism and inflammation [36]. Meng et al. demonstrated increased lipid accumulation due to overexpression of GAS5 in THP-1 foam cells. By interacting with *EZH2*, *GAS5* can prevent the expression of *ABCA1*, which was reduced by *EZH2* overexpression, resulting in decreased cholesterol efflux [37].

### 2.3. Apoptosis, Phagocytosis, Efferocytosis

A growing body of evidence suggests that apoptosis of macrophages present in lesions is closely related to atherosclerosis stages [38]. A recent study by Simion et al. demonstrated the specific presence of proatherogenic lncRNA *MAARS* in the macrophages present in plaques. *MAARS* knockdown resulted in significantly reduced areas of atherosclerotic plaques in the thoraco-abdominal aorta and aortic sinus in *Ldlr*−/− mice. It is proposed that *MAARS* may act upon the RNA-binding protein HuR one of the critical mediators of transcript stability and an apoptosis regulator. HuR nucleus-cytosolic shuttling was found to be reduced in macrophages overexpressing MAARS. Additionally, in *Ldlr*−/− mice, attenuation of interactions between *MAARS* and HuR significantly decreased pro-apoptotic markers, such as p53, p27, and caspase-3, -8, and -9, whereas anti-apoptotic markers, such as BCL2, Mcl1, and ProtA, increased in the intima of atherosclerotic lesions. *MAARS* knockdown has also shown the potential to increase macrophage clearance, or efferocytosis through decreased expression of c-Mer tyrosine kinase, which acts as a cell surface receptor and signalling molecule mediating efferocytosis [38]. Another lncRNA, *MIAT*, expresses its proatherogenic roles through several mechanisms. One of the recent studies showed that the upregulation of *MIAT* is stimulated by ox-LDL, particularly in macrophages of advanced atherosclerotic lesions in *ApoE*−/− mice [39]. In vivo *MIAT* knockdown resulted in a decreased necrotic core and increased plaque stability, primarily through increased macrophage-mediated clearance of apoptotic cells. This effect was also observed in vitro. Additionally, inhibition of efferocytosis by *MIAT* was observed, most likely targeting the miR-149-5p/CD47 axis [39]. Another important lncRNA with pro-atherogenic function is *LIPCAR*. In the study by Hu N et al. [40], overexpression of *LIPCAR* in oxLDL-treated human acute monocytic leukaemia (THP-1) cells resulted in increased lipid accumulation leading to the formation of foam cells. Additionally, the knockdown of *LIPCAR* in THP-1 cells could potentially reverse ox-LDL-mediated inhibition of cell proliferation and apoptosis. Another lncRNA that may have a role in macrophage function is *PELATON*. Namely, this lncRNA is predominantly located in the nucleus of monocytes and monocyte-derived macrophages. In one of the recent studies [41], knockdown of *PELATON* in macrophages resulted in the impairment of phagocytosis. Alterations in different phagocytosis processes were observed, such as the uptake of oxLDL particles or production of reactive oxygen species, which are closely associated with the progression of stable plaques into vulnerable or unstable phenotypes. The knockdown of *PELATON* resulted in the downregulation of CD36, one of the scavenger receptors that recognizes dying cells and oxLDL particles, demonstrating its role as a critical regulator of phagocytosis [41].

### 2.4. Inflammatory Response and Oxidative Stress

Early atherosclerosis is characterized by oxidative stress and inflammation. Macrophages play an important role in the establishment of a chronic inflammatory state seen in the development and progression of atherosclerosis. Inflammation is always accompanied by the increased production of reactive oxygen species, and subsequent oxidative damage. Several lncRNAs have been shown to regulate the inflammatory response in macrophages. A recent study by An JH et al. [42] found that the lncRNA *SNHG16* is upregulated in patients with AS and promotes inflammatory responses in THP-1 macrophages. They also discovered that overexpression of this lncRNA leads to a marked downregulation of miR-17-5p, which in turn activates the NF-κB pathway and triggers an inflammatory response in AS patients and THP-1 macrophages [42]. Another lncRNA that has the ability to activate proinflammatory macrophages via the NF-κB pathway is *MIAT* [43]. Elevated levels of *MIAT* have also been found in the sera and plasma of symptomatic patients [39,44,45]. Elevated *MIAT* levels correlated positively with serum levels of TNFα, IL-6 [45] and IL-8 [44] and were also associated with lower levels of IL-10 [44]. Consistent with this, different polymorphisms in the promoter region of *MIAT* could potentially confer a higher risk of developing myocardial infarction [46]. *NEAT1* can also contribute to atherosclerosis progression through oxidative stress and inflammatory pathways [47]. Chen et al. demonstrated that stimulation of macrophages with ox-LDL leads to upregulation of *NEAT1*. Furthermore, *NEAT1* silencing in macrophages led to the downregulation of CD36, IL-6, IL-1β, TNF-α, a decrease in foam cell formation, and suppression of reactive oxygen species (ROS) and malondialdehyde (MDA) levels through an increase in superoxide dismutase (SOD) activity. The authors speculated that these effects could be dependent on sponging of miR-128. *NEAT1* inhibition also helps to suppress inflammation through significant reduction in levels of IL6, IL1, cyclooxygenase 2 (COX2), and TNF-α levels in THP1 cells [47].

### 2.5. Endothelial Cells

#### Endothelial-to-Mesenchymal Transition

Endothelial-to-mesenchymal transition (EndMT) is the process in which endothelial cells are given mesenchymal cell properties, such as VSMCs or fibroblasts [48]. This transition is a critical step in the development and progression of atherosclerotic plaques and the subsequent transition into unstable plaques. *MALAT1* may exhibit proatherogenic effects by influencing EndMT. In human umbilical vascular endothelial cells (HUVECs) treated with ox-LDL particles, a significant decrease in endothelial markers such as CD31 and vWF and a significant increase in mesenchymal markers such as α-smooth muscle actin (α-SMA) and vimentin was observed, which is a strong indicator of ox-LDL induced EndMT. *MALAT1* silencing in HUVECs resulted in a morphologic transition and increase in endothelial markers. EndMT induced by oxidized LDL is most likely mediated by the ability of *MALAT1* to activate the Wnt/β-catenin pathway. Overexpression of *MALAT1* was found in arterial tissues of atherosclerotic mice, indicating its possible involvement and effect in the development of unstable plaques [48].

### 2.6. Migration and Proliferation

Endothelial cell migration and proliferation contribute to the progression of atherosclerosis by promoting the formation of atherosclerotic plaques. As the migrating endothelial cells penetrate the vessel wall, they initiate an inflammatory response and recruit immune cells. Several lncRNAs have been shown to influence this process. LncRNA at the INK4 locus (*ANRIL*, *CDKN2BAS*) plays an important role in atherogenesis since it is involved in almost every mechanism of atherosclerotic initiation and progression, including migration, proliferation, apoptosis of endothelial cells [49]. Activation of NF-κB by TNF-α increases the expression of *ANRIL* that forms a functional complex with YY1 and consequently upregulates IL-6 and IL-8 causing endothelial damage [50]. In addition, the expression of *ANRIL* is positively correlated with the expression of *CARD8* (a member of the caspase recruitment domain (CARD)–containing family) [51]. *CARD8* plays a significant role in endothelial activation by regulating the expression of cytokines and chemokines in endothelial cells and atherosclerotic lesions. Furthermore, *ANRIL* also increases the expression of VEGF, an angiogenic factor, which induces migration and proliferation of endothelial cells and increases vascular permeability through interaction with p300, PRC2 and miR-200b [52]. The proliferation of endothelial cells is also controlled by *NEAT1* [53]. In oxLDL-treated HUVECs, *NEAT1* knockdown promotes proliferation while suppressing apoptosis and inflammation by upregulating miR-30c-5p and reducing expression of T-cell-specific transcription factor-1 (transcription factor implicated in advances of several chronic diseases). In contrast to *ANRIL* and *NEAT1*, *lincRNA-p21* reduce atherosclerosis progression by influencing miR-221/SIRT1/Pcsk9 axis. Namely, *lincRNA-p21* acts as a sponge for mir-221 sequestering it from its target *SIRT1* gene that is involved in cell proliferation and fibrosis [54].

Laminar flow regulates a plethora of protective vascular genes, which have a significant impact on the normal morphology and functions of ECs [54]. Leisegang et al. showed that the downregulation of *MANTIS* leads to a significant alteration of the expression levels of flow-induced genes *ICAM1*, *LINC00920*, Collagen Type III Alpha 1 Chain (*COL3A1*), and Semaphorin 3A (*SEMA3A*) in HUVECs [54]. Furthermore, the authors demonstrated that overexpression of *MANTIS* in the laminar flow-induced inertial state of HUVECs inhibited transcription of ICAM1, whereas downregulation of *MANTIS* mediated nearly five-fold increased adhesion of THP-1 cells. Laminar flow conditions increased MAP-kinase-5 activity, which phosphorylates the transcription factor MEF2. Phosphorylated MEF2 activates KLF2 and KLF4, which leads to the expression of *MANTIS. MANTIS* was found to be significantly downregulated in carotid plaques. Interestingly, *MANTIS* expression levels in carotid plaques of patients on statin therapy were higher than in those not on statin therapy. These beneficial statin effects were diminished in unstable plaques [54].

### 2.7. Inflammatory Response

The inflammatory response of endothelial cells plays a critical role in the pathogenesis of atherosclerosis. Dysfunctional endothelium upregulates the expression of adhesion molecules leading to the recruitment of immune cells and progression of atherosclerosis. LncRNA *AK136714* atherogenic effects are related to its ability to induce inflammatory response in endothelial cells. *AK136714* binds to HuR, also known as ELAVL1. HuR is a widely expressed RNA-binding protein that increases the stability of TNF-α, IL-1β and IL-6 mRNA, which promotes inflammation [55]. In addition, Ming-Peng and colleagues indicated the potential significance of the *COLCA1* (colorectal cancer associated 1)/miR-371a-5p/SPP1 axis in atherosclerosis-related inflammation [56]. Stimulation of coronary endothelial vascular cells with ox-LDL particles was followed by upregulation of *COLCA1*, which led to downregulation of miR-371a-5p, and consequently to upregulation of *SPP1* (also known as osteopontin). SPP1 is a proinflammatory cytokine that stimulates the production of IFN-γ and IL-12. All these changes prevent endothelial cells from wound healing, making the vascular endothelium susceptible to plaque formation and progression [56].

### 2.8. VSMCs

#### Phenotype Switching, Proliferation and Migration

The ability of vascular smooth muscle cells (VSMCs) to switch phenotypes seems to be critical in atherosclerosis progression, as they migrate, proliferate, and contribute to intimal hyperplasia, arterial wall degeneration, and restenosis. Dysregulated VSMC functions, including proliferation and migration, play key roles in vascular remodelling and the development of atherosclerotic plaques, which are significant in arteriosclerosis. The proliferation, migration, senescence, and apoptosis of VSMCs are all influenced by the ab-errant expression of *ANRIL* [57,58,59]. *ANRIL* and *SUZ12*, a vital part of the PRC2 complex, work together to silence p15INK4, responsible for necrotic debris accumulation in the plaque [57]. On the other hand, as a result of the interaction between *ANRIL* and the chromodomain of chromobox homolog 7 (*CBX7*, a subunit of PRC1), *p16INK4*, an inhibitor of kinases responsible for cell apoptosis, and PRC1 are combined [58,59]. Both processes lead to VSMC proliferation and the development of plaque [57,58,59]. On the contrary, in the *apoE*−/− mouse model, metformin treatment resulted in the upregulation of *ANRIL*, which consequentially inhibited phenotype switching of VSMCs and the development of atherosclerotic plaques. These alterations in phenotype switching were mediated through *ANRIL’s* activation of AMP-activated protein kinase (AMPK) [60]. LncRNA *H19* may promote vulnerable plaque formation also by influencing the proliferation and migration of VSMCs. Knockdown of *H19* may induce apoptosis of VSMCs in a p53-dependent manner, thus slowing the progression of atherosclerosis [61]. The upregulation of *H19* in *apoE*−/− mice could also promote the formation of vulnerable plaques by downregulating *PKD1* via the recruitment of CTCF. Moreover, it has been shown that knockdown of *H19* leads to downregulation of MMP-2, VEGF, and p5*3* and upregulation of *TIMP*-1 expression [62]. There is growing evidence that *LIPCAR* also plays a critical role in phenotype switching and overall differentiation, proliferation, and migration of human VSMC [63]. A significant increase in the expression of the lncRNA *LIPCAR* was observed in human VSMCs treated with ox-LDL particles and PDGF-BB [63]. In addition to *LIPCAR*, these cells showed increased expression of PCNA and cyclin D2, which exhibit potent proliferative effects, whereas the expression of p21, one of the major anti-proliferative genes, was significantly decreased. Upregulation of *LIPCAR* resulted in increased expression of *MMP2* and *MMP9*, which regulate VSMC migration. Finally, this study demonstrated that upregulation of LIPCAR could potentially lead to facilitated migration of VSMCs via *CDK2*/*PCNA* upregulation [40]. LncRNA *MIAT* is involved in the activation of the ERK-ELK1-EGR1 pathway, which is a major contributor to vascular smooth muscle cell (VSMC) proliferation [44]. LncRNA *AL355711* acts as an enhancer of atherogenesis by promoting the migration of VSMCs [64]. The overexpression of *AL355711* leads to the overexpression of *ABCG1*, which in turn promotes the expression of *MMP3*. Moreover, silencing of *AL355711* inhibits the migration of VMSCs [64]. LncRNA *ZNF800* shows its atheroprotective role by inhibiting migration and proliferation of VSMCs by increasing PTEN protein expression, which leads to the inhibition of the AKT/mTOR signalling pathway and inhibition of the VSCMs proliferation [65]. LncRNA *NEXN*-*AS1* that is significantly downregulated in atherosclerotic plaques mitigates atherosclerosis by regulating the actin-binding protein NEXN, and through inhibition of the TLR4/NF-κB signalling pathway results in decreased migration of VSCMs [66].

Several lncRNAs influence VSCMs functions by acting as miRNA sponges. *H19* act as a competing endogenous RNA via interaction with let-7a miRNA, which promotes the expression of cyclin D1 and favours the proliferation of VSMCs [67]. *LINC0123*, showed a potent pro-atherosclerotic effect acting through the miR-1277-5p/KLF5 axis and causing migration and proliferation of VMSCs [68]. LncRNA *SNHG8* promotes the proliferation and migration of VSMCs by via sponging of miR-224-3p [69], while lncRNA *SNHG7*-*003* decrease proliferation, migration and invasion of VSMCs by binding miR-1306-5p/SIRT7 [70].

### 2.9. Apoptosis

Apoptosis of VSMCs is one of the critical mechanisms that leads to the destabilization of atherosclerotic plaques. In the complex of pathological molecular mechanisms of atherosclerosis, the loss of function in lncRNA *CERNA1* might play a pivotal role in plaque destabilization [71]. Despite the fact that *CERNA1* overexpression had no effect on lipid droplet accumulation, there was a significant increase in the number of VSMCs and anti-inflammatory macrophages in the plaques of the *CERNA1*-overexpressed group of *apoE*−/− mice [71]. Furthermore, *CERNA1* overexpression led to a significant decrease in MMP2/9 activity and IL6 expression levels. Increase in cell numbers and anti-inflammatory effects are most likely mediated by *CERNA1′s* ability to induce the expression of API5, which in turn inhibits VSMCs and anti-inflammatory macrophage apoptosis, which ultimately led to the stabilization of atherosclerotic plaques in the *apoE*−/− mouse model [71]. In contrast to handful of studies showing pro-atherogenic effects of *NEAT1*, overexpression of *NEAT1* inhibits VSCMs apoptosis by increasing expression of Bmal1/Clock and decreasing levels of Bax, cytochrome c, and cleaved caspa-se-3 expression [72]. LncRNA-*SNHG14* could potentially inhibit VSMCs proliferation, but induce apoptosis through its sponging of miR-19a-3p, which results in overexpression of *RORα* [73].
cells-12-01832-t001_Table 1Table 1LncRNAs implicated in the regulation of plaque stability.NameSpecies */Chromosome */Class *↑/↓Cell TypeMechanism of Action*AK136714*MM/Chr 6/?↑HUVECs↓HuR ↑mRNA of TNF-α, IL-1β and IL-6 [55]*AL355711*HS/Chr 21/?↑VSMCs↑ABCG1, MMP3 [64]
MM/Chr 6/?


*ANRIL*HS/Chr 9p21.3/AntisenseMM/Chr 4 C4/Antisense↑↑↑↑↑↑HUVECsHRECsHUVECs, HepG2VSMCsVSMCs↑IL-6 ↑IL-8 [49]↑VEGF [49]↑CARD8 [49]↓p15^INK4b^ ↓p16^INK4a^ [49] ↑AMPK [60]*CERNA1*HS/Chr 15q21.2/Intergenic↑VSMCs↑API5 [71]*CHROME*HS/Chr 2q31.2/Antisense↑THP-1 macrophages↓miR-27b, miR-33a, miR-33b, and miR-128 ↑ABCA1, OSBPL6, NPC1 [34]*COLCA1*HS/Chr 11q23.1/Antisense↑Coronary vascular ECs↓miR-371a-5p ↓SPP1 [56]*GAS5*HS/Chr 1q25.1/Antisense↑Macrophages↓miR-135a [36]MM/Chr 1 H2.1/Antisense↑THP-1-derived FC↑EZH2 ↓ABCA1 [37]*H19*HS/Chr 11p15.5/Intergenic↓VSMCs↑p-53 pathway [61]↑VSMCs↓let-7a miRNA ↑cyclin D1 [67]MM/Chr 7 F5/Intergenic↑↑ApoE^−/−^ mice plaquesMCL↓PKD1 [62]↓miR-29a ↑IGF-1 [74]*KCNQ1OT1*HS/Chr 11p15.5/Antisense↑THP-1 macrophages↓miR-452-3p ↑HDAC3 ↓ABCA1[32]MM/Chr 7 F5/Antisense↓THP-1 macrophages↑miR-137 ↓TNFAIP1[75]*LINC01123*HS/Chr 2q13/Intergenic↑VSMCs↓ miR-1277-5p, KLF5 [68]*lincRNA-p21*HS/Chr 6p21.2/Intergenic↑HAECs↓miR-221 ↑SIRT1 ↓Pcsk9 [54]MM/Chr 17 A3.3/Intergenic


*LIPCAR*HS/Mitochondria/Intergenic↑Human VSMCs↑PCNA, cyclin D2 [63]↑THP-1↑CDK2/PCNA [40]*MAARS*MM/Chr 2 C3/Sense-overlapping↑Macrophages↓HuR [38]*MALAT1*HS/Chr 11q13.1/Intergenic↑HUVECs↑Wnt/β-catenin pathway [48]MM/Chr 19 A3/Intergenic↑↑HUVECsDendritic cells↓miR-216a-5p ↑Beclin 1 [76] ↓PI3/AKT pathway [77] ↓miR-155-5p ↑NFIA [78]*MANTIS*HS/Chr 2p13.3/Intergenic↓HUVECs↓MAP-kinase-5, MEF2, KLF2 and KLF4 [79]*MeXis*MM/Chr 4 B2/Intergenic↑Macrophages↑DDX17 ↑ABCA1 [80]*MIAT*HS/Chr 22q12.1/IntergenicMM/Chr 5 F/Intergenic↑Macrophages↓miR-149-5p ↑CD47 [39]↑VSMCs↓miR-29b-3p ↑PAPPA [44]↑Human carotid artery SMCs↑ERK-ELK1-EGR1 pathway [43]↑Macrophages↑NF-κB signalling [43]*NEAT1*HS/Chr 11q13.1/IntergenicMM/Chr 19/Intergenic↓↓↓↑↑RAW264.7 macrophagesTHP-1 macrophagesHUVECsHUVECsVSMCs↓CD36, IL-6, IL-1β, TNF-α, ROS, ↑SOD [47]↓IL6, IL1, COX2, TNF-α [81]↑miR-30c-5p, ↓TCF-1 [53]↓miR-185d-5p ↑CDKN3 [82]↑Bmal1/Clock [72]*NEXN-AS1*HS/Chr 1p31.1/Antisense↑ECs, VSMCs, monocytes↑NEXN ↓TLR4/NF-κB signalling pathway [66]*PCA3*HS/Chr 9q21.2/Intronic↑ApoE^−/−^ mice-derived FCs↓miR-140-5p↑RFX7,ABCA1 [35]*PELATON*HS/Chr 20q13.13/Intergenic↓Macrophages↓CD36 [41]SNHG12HS/Chr 1p35.3/Antisense↑MacrophagesDNA damage and senescence [35]MM/Chr 4 D2.3/Intergenic*SNHG14*HS/Chr 15q11.2/Antisense↑VSMCs↓miR-19a-3p ↑RORα [73]MM/Chr 7 B5/Intergenic*SNHG16*HS/Chr 17q25.1/Sense-overlapping↑THP-1 macrophages↓miR-17-5p ↑NF-κB signaling pathway [42]MM/Chr 11 E2/Sense-overlapping


*SNHG7-003*HS/Chr 9q34.3/Antisense↑VSMCs↓miR-1306-5p [70]MM/Chr 2 A3/?


*ZNF800*HS/Chr 7q31.33/Intronic↑VSMCS↑PTEN ↓AKT/mTOR/HIF-1α signaling [74]MM/Chr 6/Intronic


*—Data were gathered using information found on the following websites: https://www.encodeproject.org/ (accessed on 15 June 2023); http://www.ensembl.org/index.html (accessed on 15 June 2023); https://www.informatics.jax.org/ (accessed on 15 June 2023); https://www.ncbi.nlm.nih.gov (accessed on 15 June 2023); https://www.genenames.org/ (accessed on 15 June 2023); https://lncipedia.org/ (accessed on 15 June 2023); https://ngdc.cncb.ac (accessed on 15 June 2023). All of the aforementioned sites were assessed in April 2023.; HS—Homo Sapiens; MM—Mus Musculus; NA—Not applicable; ↓—downregulated/decreased activity; ↑—upregulated/increased activity; ApoE−/− mouse model—mice with knocked-down expression of apolipoprotein E; CARD8—Caspase activation and recruitment domain 8; ECs—endothelial cells; FC—foam cells; HAECs—human aortic endothelial cells; HIF-1α—Hypoxia Inducible Factor 1 Subunit Alpha; HRECs—Human retinal endothelial cells; HUVECs—human umbilical cord endothelial cells; IL-6—interleukin 6; IL-8—interleukin 8; MCL—myocardial cell line; mTOR—the mammalian target of rapamycin, PTEN—Phosphatase and tensin homolog, SMCs—smooth muscle cells; THP-1—human acute monocytic leukaemia cell line; VEGF—vascular endothelial growth factor; VSMCs—vascular smooth muscle cells.


## 3. Challenges in Translational Research of lncRNAs in Atherosclerosis

As presented above, lncRNAs have a significant role in the regulation of molecular mechanisms involved in the atherosclerotic process. Different lncRNAs contribute to the control of lipid metabolism and inflammation in endothelial cells and VSMCs and participate in different stages of plaque development and progression. This implies the possibility that plaque phenotypes may differ by their lncRNA signature. Given these possibilities, there are currently many attempts to use lncRNAs as potential drug targets and/or biomarkers that could be translated from bench to bedside. However, the functions of several lncRNAs in atherosclerosis remain controversial. This may be, at least to some extent, a consequence of the different models and methodologies used. Hence, we would like to draw the reader’s attention to possible challenges that should be taken under consideration when planning translational research of lncRNAs in atherosclerosis.

Studies aiming to functionally analyse specific lncRNAs often require appropriate animal models. Currently, a handful of animal models are available for the investigation of atherosclerosis. However, no model is an ideal representation of human ASCVD as each has various advantages and limitations regarding lipoprotein metabolism, atherosclerosis development, localization, and histological characteristics of the atherosclerotic plaque, resulting in an ASCVD outcome different from those observed in humans.

The most extensively used murine models for atherosclerosis translational research are LDL receptor-deficient mice (*Ldlr*−/− mice) and apolipoprotein E-deficient mice (*ApoΕ*−/− mice) [83]. Although these models can develop a proatherogenic lipid profile and bypass the natural atheroresistance of wild-type mice, allowing efficient progression of atherosclerosis, the resulting plaques do not correspond to those seen in human [84]. On the other hand, the Watanabe heritable hyperlipidaemic and Golden Syrian hamster can spontaneously develop human-like plaques located in coronary arteries that can progress to plaque rupture and thrombotic occlusion causing myocardial infarction [85,86,87]. Larger animal models such as pigs, including transgenic minipigs (such as Wisconsin Miniature Swine—WMS, Rapacz familial hypercholesterolemic—RFH, etc.), are particularly well suited for studying the heterogeneous interaction of haemodynamics and vascular endothelial responses because of their size and cardiovascular anatomy, which is similar to humans [88]. They can spontaneously develop atherosclerosis, exhibit human-like lipoprotein profiles, and atherosclerotic plaques similar in complexity to human coronary lesions. For this reason, the swine model is valuable for studying molecular mechanisms of plaque development with direct translatability to humans. Pigs also have a closer genetic resemblance to humans than rodents and housing and handling are easier compared to non-human primates [88]. However, the data from animal models exploring lncRNAs’ heterogeneity and structural/functional resemblance to humans are currently sparse and impose additional challenges in translational research (Figure 2).

LncRNAs are not very well conserved between rodents and primates. In fact, close to 80% of lncRNAs in primates are not conserved in mice [89,90]. Primate-specific lncRNAs are not positionally conserved in rodents most of the time, meaning that at genomic level 5′ and 3′ neighbouring protein-coding genes are different. This limits studies aiming to explore lncRNAs cis-regulatory functions. In addition, those lncRNAs that are positionally conserved between primates and mice have very low or no expression in mice. Importantly, lncRNAs are known to form secondary structures that might be more significant for their functionality than primary sequence [91]. Thus, lack of sequence homology does not necessarily translate to the lack of functional homology. Since lncRNAs specific to primates often express specific and very important roles in the regulation of cellular functions, there is a great deal of interest in overcoming the abovementioned challenges and developing appropriate animal models. A transgene approach can be used for those lncRNAs not present in rodents that target protein-coding genes well conserved between primates and rodents. This approach was used to overexpress primate-specific lncRNA in mouse brains that led to a human-like phenotype of neuron development [92]. In addition, Van Keurren et al. used bacterial artificial chromosome transgene mice to transfer not only specific lncRNA, but also genomic regulatory elements [93]. With this approach, it was possible to achieve physiological expression levels of lncRNAs, but also tissue expression patterns as endogenous lncRNAs in vivo. Another approach that could be used in order to explore lncRNAs functions under more relevant conditions includes transplantation of human cells that express specific lncRNAs into rodents with immune deficiency. This strategy has been successfully used to study the role of human lncRNAs in cancer development and progression [94], and protein-coding genes in the function of liver and adipose tissue [95,96].

In contrast to mRNAs, lncRNAs are not very abundant. Their expression levels are in average 10 times lower in comparison to mRNAs [97]. This, of course, represents additional challenge in lncRNA investigation. In the atherosclerosis research in particular, the situation is even more challenging considering that the amount of tissue available for RNA extraction is often very low and hardly accessible. Since low-abundant targets are very challenging to accurately measure, in order to obtain reliable results for differential gene expressions it is recommended to use a larger sample size and much larger sequencing depth compared to mRNA-focused experiments [98]. Furthermore, another important step vital for the success of lncRNA investigation is adequate library preparation. One of the methodological solutions that can be used to increase lncRNAs’ detection sensitivity is RNA capture sequencing [99]. In this method, biotinylated probes designed to capture RNA of interest (in this case lncRNA) are used. In this way, overabundant mRNAs are removed from the sequencing library, enabling increased sequencing coverage of captured lncRNAs. However, this approach is applicable for targeted analysis. If the lncRNAs of interest are poly-adenylated, then so-called polyA selection can be used. However, since many lncRNAs do not contain the polyA sequence, this approach leads to loss of valuable information. One of the most comprehensive approaches to generating RNA sequencing libraries is based on the depletion of ribosomal RNAs. This method can quantify both linear and circular RNA, but it requires a higher sequencing depth compared to polyA-selection due to the increased library complexity. Total RNA sequencing can capture precursor transcripts with introns, which can be used to differentiate transcriptional and post-transcriptional regulation when combined with exon expression levels. This type of analysis may provide valuable insights into the mode of action and target genes of non-coding RNAs.

Additional challenges are related to the incomplete annotation of lncRNAs, with many lncRNAs lacking functional annotations, making it difficult to assign biological functions to these transcripts [100]. In addition, lncRNAs often undergo alternative splicing, resulting in a variety of transcript isoforms with different, sometimes even opposite functions. However, identifying and quantifying these isoforms accurately is challenging, as some isoforms may be expressed at very low levels [101].

LncRNAs are present in various biological fluids, which makes them potential candidates for biomarker research. However, lncRNAs levels in blood are very low and their quantification requires additional consideration. This has been extensively discussed elsewhere [102].

## 4. Current Advancements and Future Perspectives

Although the lncRNA research field has grown considerably during the last decade, there are still many unknowns and needs for improvement. The discovery of novel lncRNAs and analysis of their roles in atherosclerosis could help us uncover novel disease phenotypes. RNA sequencing of atherosclerotic plaques has already shown great potential for patient stratification [103]. With deeper sequencing, we could achieve even better insights into lncRNAs’ signature heterogeneity between different plaque types. Discovering novel lncRNAs that control biological processes causal for atherogenesis can drive novel therapeutic strategies. With the latest advancement in the field and the development of RNA-based therapies for CVD, it will certainly be possible to target specific lncRNAs in different patient subgroups, thus tailoring more patient-oriented therapies. In particular, the latest data showed that around 60% of lncRNAs are tissue-specific, which makes them exceptionally good tissue-specific drug targets that can be used for more precise treatment with less off-target effects [104].

Several lncRNAs have been associated with controversial roles in atherogenesis, such as *MALAT1*, *NEAT1*, *H19*, and *ANRIL*. The subcellular distribution of lncRNAs, such as nuclear or cytoplasmic localization, can influence their regulatory mechanisms and interactions with target genes. Additionally, the cellular origin of these lncRNAs and the presence of different isoforms might contribute to the ambiguous results observed in various studies. Notably, *ANRIL* has several alternatively spliced isoforms with distinct functions. Some isoforms of *ANRIL* have been reported to promote atherosclerosis by regulating the expression of genes involved in inflammation and lipid metabolism. Conversely, other isoforms have been suggested to exert protective effects by modulating cell proliferation and senescence. Certain isoforms of *H19* have been associated with pro-atherogenic effects, promoting inflammation and smooth muscle cell proliferation, while other isoforms have been linked to anti-atherogenic properties, such as inhibiting vascular cell adhesion molecules and reducing endothelial inflammation. Similarly, different isoforms of *MALAT1* have shown divergent effects in atherosclerosis, with some isoforms promoting smooth muscle cell proliferation and migration, while others have been implicated in inhibiting vascular calcification and inflammation. The presence of multiple isoforms with opposing functions underscores the complexity of lncRNA biology and highlights the importance of investigating isoform-specific effects.

This conventional pattern of lncRNA examination has provided valuable insights into the cytoplasmic potential of lncRNAs to regulate cellular processes. Most studies in the field have focused on the cytoplasmic functions of lncRNAs acting as miRNA sponges. However, a significant proportion of lncRNAs are also localized in the nucleus, suggesting their involvement in nuclear gene regulation related to plaque formation. Thus, to comprehensively unravel the functional mechanisms of lncRNAs in the context of plaque instability, it is imperative to consider and explore lncRNA functions in the nucleus. Future studies should elucidate the specific roles of nuclear lncRNAs in regulating gene expression and cellular processes associated with plaque instability, providing a comprehensive understanding of their contributions to the pathogenesis of atherosclerosis.

The availability of new technologies such as single-cell RNA sequencing and spatial RNA sequencing can deliver an additional layer of information very relevant to plaque investigation. Recent efforts to use single-cell RNA sequencing gave deeper insights into the cellular complexity of atherosclerotic plaques and reveals the cellular origins of specific lncRNAs in animals and humans. Cochain et al. demonstrate the power of single-cell RNA sequencing technology in dissecting the transcriptional landscape and heterogeneity of aortic macrophages in murine atherosclerosis [105]. The study provides valuable insights into the underlying mechanisms of atherosclerosis and may contribute to the development of targeted therapies aimed at modulating specific macrophage populations to mitigate plaque progression and enhance plaque stability. Importantly, the study also provides valuable information regarding the expression of several lncRNAs discussed in this review across different immune cell types. *NEAT1* was observed in monocyte-derived dendritic cells, inflammatory macrophages, monocytes, and neutrophils, while *MALAT1* was predominantly expressed in T cells and resident-like macrophages. Furthermore, *KCNQ1OT1* exhibited selective expression in T cells. Additionally, the study provides insights into the fold changes of these lncRNAs in response to treatment, further enhancing our under-standing of their potential functional roles in atherosclerosis. Depuydt et al. provided another important insight into lncRNA distribution, this time in humans [106].

They used single-cell RNA sequencing on human carotid atherosclerotic plaques and identified 14 different cell populations. Interestingly, their dataset excluded *MALAT1*, *KCNQ1OT1*, and mitochondrial genes (due to constitutive expression patterns), but found several lncRNAs mentioned in this review to be specific for certain cell populations. For example, *MIAT* was found only in T cells, *GAS5* in T and B cells, and *SNHG7* in endothelial cells. In addition, *NEAT1*, which is considered to be widely expressed, was present only in T and myeloid cells, but not in endothelial cells, VSMCs, nor B cells. *ANRIL* was expressed only in a mixed population of cells, which the authors described as not showing a clear cell type-defining expression profile but which had similar gene expression levels as other clusters and seemed to mainly contain apoptotic myeloid and T cells. However, it should be noted that some of the discussed lncRNAs are not very abundant, which raises the question of the relevance of their influence on atherosclerotic processes. Leveraging the data obtained from the Tabula Sapiens consortium [107], we present in Figure 3 an analysis of the expression patterns and levels of lncRNAs in the human vasculature. Specifically, we focus on the lncRNAs that have been discussed in this review and are accessible in the database. Based on the data it is clear that there is a vast difference in levels of gene expression between different lncRNAs, as well as their localisation in various cell types. These findings highlight the importance of considering the specific cell types, their heterogeneity, and their abundance when examining the expression of lncRNAs in atherosclerosis. The available single-cell RNA sequencing dataset can be utilized to determine the precise cell types expressing specific lncRNAs and gain insights into their functional roles within specific cellular contexts. In contrast to single-cell RNA sequencing, spatial RNA sequencing enables the identification of cell subtypes and states within a tissue and the mapping of cell-to-cell communication and signalling pathways within a tissue. It is a powerful tool for understanding the complex role of lncRNA-related interactions between cells in a tissue, and it has the potential to provide new insights into the disease mechanisms.

The epitranscriptomic field of research that aims to uncover how nucleotide modifications influence functions of RNAs is rapidly growing. One of the most extensively studied RNA modifications is N6-methyladenosine (m^6^A). It has been showed that this modification apart from being functionally relevant for many mRNAs can also influence lncRNAs. For example, the action of X-inactive specific transcript (*XIST*) upon its target genes is dependent on m^6^A modification and binding of m^6^A “reader” RNA binding motif protein 15 (*RBM15*) [108]. M^6^A modification in *MALAT1* can lead to conformational changes that increase its interaction with specific RNA binding proteins [109]. Some evidence suggests that targeting m^6^A modification on lncRNA *ZFAS1* could lead to amelioration of atherosclerosis and can be used as a possible clinical treatment [110]. However, current knowledge on the relevance of these mechanisms in atherosclerosis is very limited, and it should be further explored. The technological breakthrough developed by Oxford Nanopore allows direct RNA sequencing and provides information not only about the sequence of RNA molecules but also data about the presence of specific RNA modifications. At the moment, there is great interest in developing and validating protocols using this technology to investigate RNA modifications, and their relevance in the function of lncRNAs.

## 5. Conclusions

In order to go further and get more comprehensive understanding of how lncRNAs interact with other levels of cellular complexity during plaque genesis and progression, a multi-omic approach can be used. In multi-omic studies, transcriptomic data are merged with genomic, epigenomic, proteomic, and metabolomic data into multimodal information that can reveal novel interactions between different factors. Thus, novel upstream and downstream signalling partners of specific lncRNAs can be recognized and further studied in a more targeted manner. Since multi-omic studies tend to produce an overwhelming amount of data, different machine learning and artificial intelligence algorithms are used for data integration. This approach is just being implemented in ASCVD research and requires experts from various fields to work together on protocols and standards needed for meaningful and reproducible data. Such international initiatives are COST action CA21153 AtheroNET (Network for implementing multi-omics approaches in atherosclerotic cardiovascular disease prevention and research https://www.cost.eu/actions/CA21153 (accessed on 15 June 2023)) and HORIZON-MSCA-2021-SE-01-01—MSCA Staff Exchanges 2021 CardioSCOPE. These initiatives integrate different experts from basic scientists, clinicians, and cardiologists to bioinformaticians to foster collaborative research and bridge the current challenges in the field.

## Figures and Tables

**Figure 1 cells-12-01832-f001:**
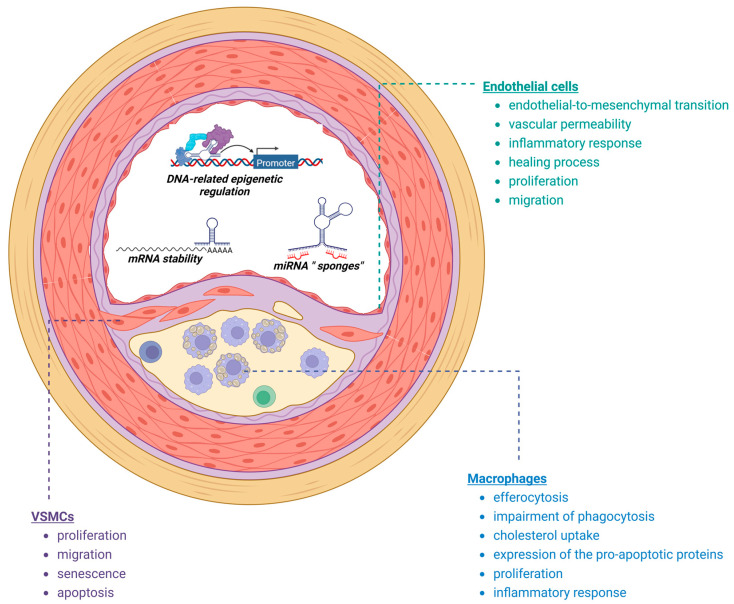
Processes implicated in plaque development and destabilization regulated by lncRNAs (created with Biorender).

**Figure 2 cells-12-01832-f002:**
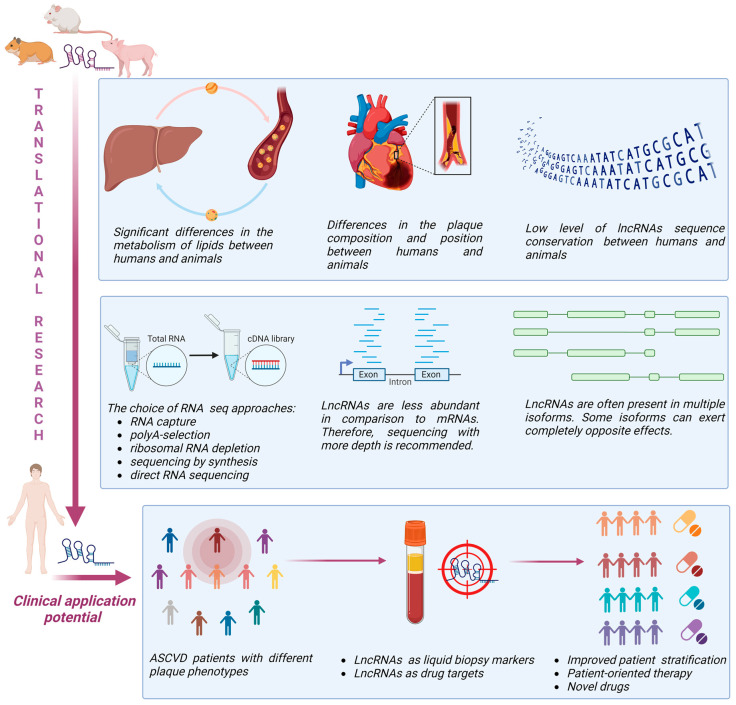
Challenges and perspectives of lncRNA translational research in ASCVD (created with Biorender).

**Figure 3 cells-12-01832-f003:**
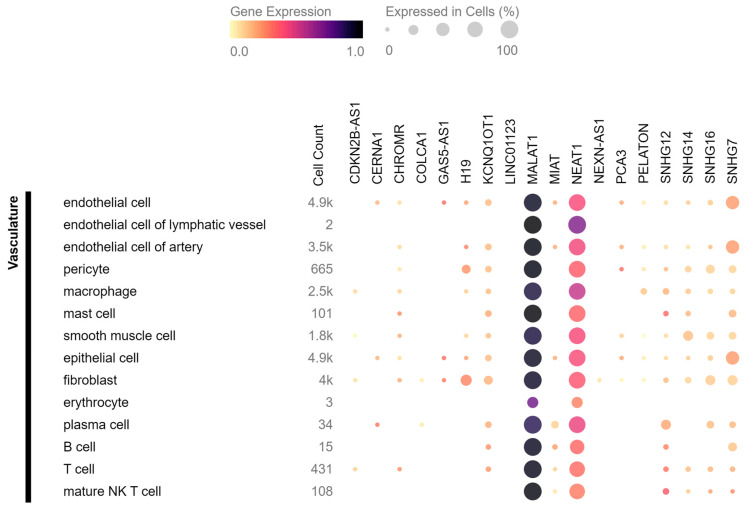
The distribution of lncRNAs related to plaque stability in human vasculature based on the data from Tabula Sapiens consortium. The colour intensity in the figure represents the varying levels of gene expression, with darker colours indicating higher expression. Additionally, the size of the circles in the figure corresponds to the percentage of cells within a specific cell type expressing the transcript of interest.

## Data Availability

Not applicable.

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
