# Peer review of "LncRNAs as Regulators of Atherosclerotic Plaque Stability"

_cells, 2023, doi:10.3390/cells12141832_

Round 1
Reviewer 1 Report (Previous Reviewer 1)
I have no more questions.
Author Response
Response to the Reviewer Number 1:
R1: I have no more questions.
Response: Thank you for your previous efforts to improve the submitted manuscript.
Reviewer 2 Report (Previous Reviewer 2)
I am not an expert in atherosclerotic cardiovascular diseases, and I am not qualified to assess the significance of this review in the atherosclerotic cardiovascular diseases field. From the perspective of RNA, this version is much better than the previous version.
Minor concerns:
1: There are still many misspellings in the manuscript. For instance, “localisation” should be “localization”. “m6A” should be “m6A”.
2: The authors should pay more attention to the name of RNA and gene. For instance, RNA “NEAT1” should be “NEAT1”.
Author Response
Response to Reviewer Number 2:
R2: I am not an expert in atherosclerotic cardiovascular diseases, and I am not qualified to assess the significance of this review in the atherosclerotic cardiovascular diseases field. From the perspective of RNA, this version is much better than the previous version.
Response: Thank you so much for your kind comments.
R2: 1: There are still many misspellings in the manuscript. For instance, “localisation” should be “localization”. “m6A” should be “m6A”.
Response: Thank you so much for your suggestion. We have run under spell check and found several more mistakes that we corrected. We have opted for the British version of English spell check, hence certain words are spelled with the "s" instead of "z" (for example, localisation, analyse, etc.) or double "l" in some words (for example, signalling, remodelling etc.) We have used uniformly used the British version throughout the text.
R2: 2: The authors should pay more attention to the name of RNA and gene. For instance, RNA “NEAT1” should be “NEAT1”.
Response: Thank you so much for your suggestion. We have corrected the gene abbreviations throughout the text accordingly.
This manuscript is a resubmission of an earlier submission. The following is a list of the peer review reports and author responses from that submission.
Round 1
Reviewer 1 Report
Cells-2373315 comments:
Atherosclerotic cardiovascular diseases (ASCVD) are still one of the leading 14 causes of death worldwide. In this manuscript, Petković et al summarized role of long noncoding RNAs (lncRNAs) in pro-atherogenic and anti-atherogenic categories. The review updated the regulatory function of lncRNAs on plaque development, including immune response, lipid metabolism, extracellular matrix remodeling, endothelial cell function, and vascular smooth muscle function. Table 1 grouped pro-atherogenic and anti-atherogenic lncRNAs in terms of their targets and related pathways.
This review manuscript is well written. It has updated current knowledge on lncRNAs and plaque development. However, it would be useful to add a short section to comment the shortcomings of existing studies. For example, most lncRNAs are located primarily in the nucleus. Thus, their functions, if any, may be involved in the nuclear regulation of genes related to plaque formation. On the contrary, most studies follow a conventional pattern to examine the sponge role of microRNAs in the cytoplasm. Thus, future studies need to focus on the role of these lncRNAs in the nucleus.
Reviewer 2 Report
The topic of this manuscript titled “LncRNAs as regulators of atherosclerotic plaque stability” is appropriate for this section. However, I do not think that the review is outstanding from the perspective of RNA biology as it lacks novelty and a constructive summary.
Concerns:
1. Several similar reviews have already been published (ref 22, 23, 29), and compared to these, this manuscript mainly summarizes the involvement or relationship of various lncRNAs with atherosclerotic plaque stability, without offering much advancement. The review seems to be a patchwork of conclusions from different articles, and while this is one way to write a review, readers may get lost in the numerous lncRNAs and not learn much from it. Given that the authors have comprehensively summarized many references, I suggest that they reorganize and rewrite the review, using different organizing principles such as by expression specificity of lncRNAs, by different stages of disease, or by different working mechanisms of lncRNAs to provide a more conceptual summary.
2. Several lncRNAs mentioned in this manuscript, such as NEAT1 and MALAT1, are widely or constitutively expressed in different tissues. It would be better to distinguish between tissue-specific and constitutively expressed lncRNAs and focus on some tissue-specific, abundant, and biologically important lncRNAs, rather than including all of them. Many of the lncRNAs have a very low level of expression, and the conclusions drawn from studies on such lncRNAs may not be reliable. Therefore, the authors should be careful when citing references.
3. The manuscript needs to pay more attention to the accuracy of the statements made. I noticed several inaccurate statements in the manuscript, such as the statement " LncRNAs are transcribed into transcripts longer than 200 nt," which does not make sense.
4. The manuscript describes both pro-atherogenic lncRNAs and anti-atherogenic lncRNAs, including NEAT1 and MALAT1, which are constitutively expressed in both groups. It is unclear whether this is because the function of NEAT1 and MALAT1 is still in debate or due to some other reason. The authors should clarify this point.
5. There is a misspelling in Figure 2. "Low level of lncNRAs sequence" should be "Low level of lncRNAs sequence".
Reviewer 3 Report
In this review, Petkovic et al has summarized the pro- and anti-atherogenic functions of 27 long noncoding RNAs based on published literature. This is a comprehensive overview of the field, and provides a useful reference resource for lncRNA researchers. I don’t have any comments on the contents of the review, but would like the authors to consider the following points to augment the utility of this review:
1. For Table 1, add the chromosomal location of the lncRNAs, the species (human/mouse etc.) and annotate the lncRNA as intergenic, intronic, etc.
2. Recently, several single cell RNA sequencing studies have been performed to generate atherosclerosis-specific cell atlas in mice and humans to elucidate the cellular heterogeneity associated with this disease which may help further refine our understanding of pathophysiology (e.g. Cochain et al. Circulation Research. 2018;122:1661–1674. doi:10.1161/circresaha.117.31
for murine atherosclerosis ; Depuydt et al., Circ. Res. 127 (11), 1437–1455. doi:10.1161/circresaha.120.316770 for human atherosclerosis). The authors could check for the expression of the reviewed lncRNAs in one or more of these published datasets to determine the specific cell-type(s) expressing them. I think this will significantly add value to this review.
Please check the manuscript for typographical errors. For example, on line 409, hypothesized has been mistyped as hypnotized.